# Tigecycline Interferes with Fibrinogen Polymerization Independent of Peripheral Interactions with the Coagulation System

**DOI:** 10.3390/antibiotics9020084

**Published:** 2020-02-14

**Authors:** Anna Brandtner, Mirjam Bachler, Dietmar Fries, Martin Hermann, Jacqueline Ruehlicke, Vilmos Fux, Andrea Griesmacher, Christian Niederwanger, Tobias Hell, Benedikt Treml

**Affiliations:** 1Laboratory for Inflammation Research, Department for Internal Medicine, Division of Intensive Care and Emergency Medicine, Medical University Innsbruck, 6020 Innsbruck, Austria; 2Institute for Sports, Alpine Medicine and Health Tourism, Private University for Health Sciences, Medical Informatics and Technology GmbH, 6060 Hall in Tirol, Austria; 3Department of General and Surgical Intensive Care, Medical University Innsbruck, 6020 Innsbruck, Austria; 4Central Institute of Medical and Chemical Laboratory Diagnostics, University Hospitals of Innsbruck, Innsbruck 6020, Austria; 5Department for Pediatrics, Pediatrics I, Intensive Care Unit, Medical University Innsbruck, 6020 Innsbruck, Austria; 6Department of Mathematics, Faculty of Mathematics, Computer Science and Physics, University Innsbruck, 6020 Innsbruck, Austria

**Keywords:** tigecycline, fibrinogen, bleeding, real time live confocal analysis, hepatocyte cell culture

## Abstract

Tigecycline offers broad anti-bacterial coverage for critically ill patients with complicated infections. A described but less researched side effect is coagulopathy. The aim of this study was to test whether tigecycline interferes with fibrinogen polymerization by peripheral interactions. To study the effect of unmetabolized tigecycline, plasma of healthy volunteers were spiked with increasing concentrations of tigecycline. In a second experimental leg, immortalized human liver cells (HepG2) were treated with the same concentrations to test an inhibitory effect of hepatic tigecycline metabolites. Using standard coagulation tests, only the activated thromboplastin time in humane plasma was prolonged with increasing concentrations of tigecycline. Visualization of the fibrin network using confocal live microscopy demonstrated a qualitative difference in tigecycline treated experiments. Thrombelastometry and standard coagulation tests did not indicate an impairment of coagulation. Although the discrepancy between functional and immunologic fibrinogen levels increased in cell culture assays with tigecycline concentration, fibrinogen levels in spiked plasma samples did not show significant differences determined by functional versus immunologic methods. In our in vitro study, we excluded a direct effect of tigecycline in increasing concentrations on blood coagulation in healthy adults. Furthermore, we demonstrated a rapid loss of mitochondrial activity in hepatic cells with supra-therapeutic tigecycline dosages.

## 1. Introduction

Of the expanded broad spectrum glycylcyclines, the substance tigecycline (tygacil), approved by the Food and Drug Administration (FDA) in 2005 and by the European Medical Association (EMA) in 2006, respectively, is indicated for complicated skin, soft tissue infections and abdominal infections in adults and infants older than eight years [1,2]. A described side effect during tigecycline therapy is coagulopathy. Three small-scale studies reported effects of tigecycline on coagulation [3,4,5]. A clinical trial observed decreased fibrinogen levels as assessed by the classical Clauss method during high-dose treatment with tigecycline in a group of 45 intensive care patients [4]. Recently, another study showed a decrease of fibrinogen in 19 out of 20 patients with severe infections treated with glycylcycline. This occurred either with the recommended dose (loading dose of 100 mg, followed by 50 mg twice per day intravenous (iv.)) and during high dose (2 × 100 mg iv. per day) treatment with clinical bleeding in six patients [5]. Hakeam et al. reported decreased fibrinogen levels during tigecycline treatment as compared to imipenem–cilastatin after cytoreductive surgery and hyperthermic intraperitoneal chemotherapy with peritoneal carcinomatosis [6]. In a retrospective analysis Leng et al. demonstrated decreased fibrinogen levels in 50 hospitalized patients with tigecycline treatment in China [3]. Additionally, eight case reports exist in regard to an effect of tigecycline on coagulation [7,8,9,10,11,12,13,14]. In reflection to these observations, the manufacturer recommends discontinuing tigecycline, if fibrinogen levels decrease below 100 mg/dL. However, it is one of few available antibiotics active against key multi-drug resistant pathogens and its use as second line antibiotic is limited to patients with no alternative suitable treatment options [15].

The origin of the decrease of fibrinogen remains elusive. Hypofibrinogenaemia concomitant to tigecycline treatment in the mentioned studies was not accompanied by a worsening of liver parameters. Thus, impaired hepatic synthesis of coagulation factors, such as fibrinogen, due to the medication is unlikely in physiological doses. Tigecycline is excreted majorily unmetabolized, but trace amounts of metabolites are generated by the liver. The major modifications include glucuronidation and amide hydrolysis. These metabolites are usually pharmacologically inactive and represent less than 10 percent of excreted tigecycline [16]. However, tigecycline was shown to interact with an atypical, non-linear binding to plasma proteins, which is dependent on a complex interaction with divalent metal ions such as calcium ions. A potential functional effect of tigecycline and its metabolites on plasma proteins, such as fibrinogen and the polymerization to fibrin networks, thus cannot be excluded.

We hypothesize that tigecycline interacts with fibrinogen, affecting the functionality of fibrinogen and alter associated laboratory measurements of coagulation parameters. The Clauss assay is a functional assay based on the determination of time for fibrin clot formation. For other drugs like starches interferences with this turbidimetric method of fibrinogen measurement are reported [17,18]. In contrast, immunological fibrinogen assays measure protein concentration rather than functional activity. However, these fibrinogen antigen concentrations correlate well with fibrinogen functional activity except in cases of dysfibrinogenemia [19,20].

In the presented study in vitro tests are used to investigate an association of tigecycline treatment and a qualitative or quantitative dysfibrinogenaemia in peripheral blood of healthy volunteers. We employ an in vitro spiking assay to globally test dose dependent peripheral interactions of tigecycline with the coagulation system, determined by functional and immunologic fibrinogen tests, as well as thrombelastometric techniques, which are an accurate method for detection of fibrin polymerization disturbances [17]. In a second experimental leg, we apply a cell model of hepatocytes to test if the most common metabolic modifications of tigecycline alter fibrinogen functionality.

## 2. Results

Plasma samples of 14 healthy subjects were obtained to perform the planned experiments. One subject had to be excluded due to missing rotational thrombelastometry (ROTEM) measurements. In the final analysis blood samples of three males and ten females, aged 21–39 years old were included.

Demographics as well as baseline laboratory parameters are shown in Table 1.

### 2.1. Influence of Increasing Concentrations of Tigecycline on Standard Coagulation Tests

Standard coagulation parameters (i.e., activated partial thromboplastin time, Quick, international normalized ratio, Fibrinogen by Clauss, Fibrinogen antigen, Thrombin time, Batroxobin time, coagulation factor XIII (F.XIII) and F.XIII immunologic) remained nearly unchanged after addition of increasing concentrations of tigecycline (Table 2). A trend to a decreased activated partial thromboplastin time could be observed only at supraphysiologic concentrations (i.e., about hundred-fold) as compared to baseline activated partial thromboplastin time (aPTT).

### 2.2. Influence of Tigecycline on ROTEM^®^ Measurements

Parameters in rotational thromboelastometry remained nearly unchanged in all three test settings of activated intrinsic and extrinsic pathway (INTEM and EXTEM), as well as fibrinogen dependent test read outs (FIBTEM). Differences between treatment groups were not shown to display systematic trends as determined by one-way ANOVA (Table 3).

### 2.3. Influence of Tigecycline on Fibrin Polymerization

In cell culture experiments, supra physiologic doses of tigecycline resulted in a rapid loss of viable cells within 72 h after treatment start (dimethyl thiazolyl diphenyl tetrazolium bromide (MTT) activity—67.6% ± 3.79% compared to untreated, *p* < 0.001, *n* = 3), while physiologic doses had no effect on viability as measured by MTT activity (−31.72% ± 2.50% compared to untreated, *p* = 0.39, *n* = 3, Figure 1).

Confocal microscopy with the immortalized human liver cells (HepG2) supernatant blended with an equal volume of citrated platelet-free plasma (PFP) from a healthy volunteer was performed to evaluate the structure of the fibrin network after incubating different concentrations of tigecycline for a maximum of ten days. The amount of fibrinogen directly produced by cultured hepatocytes were beneath the measurement limits of both the Clauss method and the immunologic fibrinogen detection, therefore supernatants of the cell culture experiments were blended with plasma from a healthy volunteer in a plasma-exchange approach [21]. In plasma exchange experiments we found qualitative differences, with a more filigree appearance of the fibrin networks with increasing concentrations of tigecycline (Figure 2).

These qualitative differences could not be detected in quantitative measurements of fibrinogen by Clauss or by immunologic detection of fibrinogen antigen in plasma samples (Table 2) or the plasma exchange experiments with supernatants of cell culture in-vitro assays (Figure 3A), despite a lower number of active hepatocytes was present in experiments with 106 µg/mL tigecycline, as determined by the MTT-assay. However, a trend indicating an increasing discrepancy between the two methods of fibrinogen quantification with higher concentrations of tigecycline was observed in measurements including conditioned media from tigecycline-treated HepG2 (Figure 3B).

## 3. Discussion

In this in vitro-experiment we sought to elucidate the cause of an erratic fibrinogen decrease in critical ill patients during tigecycline treatment. We observed small changes in fibrinogen levels und coagulation parameters after addition of supra-therapeutic tigecycline doses to the blood of healthy adults. These minor changes are of little clinical relevance, as clinical overt bleeding in critical care is mostly linked to gross fibrinogen decreases. Moreover, the fibrinogen decrease in critical ill patients tends to occur only after several days of tigecycline start [3,4]. We conclude that the blood of our healthy subjects lacks either the causative agent of the fibrinogen decrease observed in critical ill patients under tigecycline treatment or a long enough exposure to tigecycline. Moreover, patients experiencing a decreased fibrinogen under tigecycline treatment could either be related to an increased consumption or a (hepatic) synthesis problem. From a clinical point of view it is crucial if administration of fibrinogen concentrate could further fuel the underlying mechanism in case of an increased consumption. Sabanis et al. hypothesized a synthesis issue by decrease of gut bacteria during tigecycline treatment leading to an accordingly depletion of vitamin K2-dependent coagulation factors [11]. Here, we demonstrated a rapid decrease of hepatic cells viability after addition of supra-therapeutic tigecycline dosages. However, this decrease was not reflected in functional fibrinogen measurements in plasma exchange experiments. At this step, we did not use the hepatoprotective drugs like vitamine C or L-carnithin, but we hypothesize that high-dose treatment could have an effect on liver function in vivo.

A major limit in interpreting the current and sparse data on coagulopathy under tigecycline treatment are the different drug dosages used in the different trials or case reports. One could hypothesize that high-dose treatment with tigecycline is a possible risk factor for hypofibrinogenemia which is supported by the observations of Routsi et al. who found high doses of this drug leading to hypofibrinogenemia [4]. Moreover, Zhang et al. described a dose dependent decrease of fibrinogen levels [5].

The manufacturer recommends a loading dose of 100 mg, followed by 50 mg twice per day iv. for adults. One study administered mainly the recommended 2 × 50 mg iv. per day with only 5 patients receiving a higher dose [5]. Another study used higher doses (2 × 100 mg iv. in 39 patients) and another 6 patients receiving 75 mg twice a day [4]. However, patient weight was reported in neither study. Therefore, alteration of coagulation could even be due to different serum concentrations, which is especially true for critical ill patients with changing fluid balance over the course of illness. Another factor could be continuous renal replacement therapy (CRRT) which is often required in critical ill patients. However, CRRT contributes only to a minor part of tigecycline elimination [22].

Possible limitations are the small sample size and young age of the healthy volunteers. Increasing age has been shown to increase the risk of thrombosis and is an independent risk factor for cardiovascular disease [23]. On the other hand, Zhang et al. found no difference in the magnitude of fibrinogen decrease in patients being younger or older than the age of 65 [5].

In summary, our data support a possible effect of tigecycline on qualitative characteristics of the fibrin network. As the clinical significance of these qualitative changes are not assessable to date, the benefit of treatment with tigecycline in patients with underlying coagulopathy or being at risk for bleeding should be carefully weighed as the possible manifestation of a clinically relevant coagulopathy cannot be excluded.

To summarize, in vitro standard coagulation parameters remained unchanged after addition of tigecycline in increasing doses to blood of young and healthy adults. Qualitative differences could be observed regarding the architecture of the fibrin networks with increasing doses of tigecycline, but these changes had no significant effect on clot stability parameters. Peripheral interactions of tigecycline on fibrin polymerization therefore might not be the reason of coagulopathy in critically ill patients during tigecycline treatment.

## 4. Materials and Methods

### 4.1. Healthy Volunteers

Healthy adult volunteers were included into the study after providing informed consent to the study procedures and a medical check especially regarding the medical history of coagulopathies, concomitant medication interfering with coagulation parameters or any kind of liver diseases. As such the exclusion criteria comprised intake of any anticoagulant medications, the presence of any kind of haemophilia or acquired and hereditary coagulation disorders, any kind of liver disease, pregnancy, as well as the intake of medications possibly interfering with the results of the study (i.e., Aspirin, other platelet inhibitors). Within one month prior to testing several participants took low dose of nonsteroidal anti-inflammatory drugs (*n* = 5), oral contraceptives (*n* = 2), a beta-blocker (*n* = 1), a retinoid (*n* = 1), and an inhalation aerosol that contained a combination of a beta–2 sympathomimetic and a vagolytic drug (*n* = 1). The procedure was approved by the Ethics Committee of the Medical University of Innsbruck (#1199/2017).

### 4.2. Preparation of Blood Samples

This laboratory study was conducted in Innsbruck. From each healthy adult blood samples were drawn. One citrate blood sample (Vacutainer, Sarstedt, Nümbrecht, Germany) served as baseline for ROTEM^®^ and real-time live confocal microscopy, respectively. Another four citrate blood samples were spiked with logarithmically increasing concentrations of tigecycline. Thereafter fibrin polymerization was measured with ROTEM^®^ and visualized by real-time live confocal imaging. Two Ethylenediaminetetracetic acid (EDTA)-tubes (Sarstedt, Nümbrecht, Germany) were used for blood cell count and baseline thrombelastometric tests. For each spiking step, stock solutions were prepared. A 50 mg Tygacil^®^ vial (Pfizer, Kent, United Kingdom) was dissolved in 16.3 mL of 0.9% sodium chloride injection (Braun, Melsungen, Germany). As Tygacil^®^ vials contain a 6% overage, 16.3 mL equal a concentration of 3286 μg/mL. This first solution was further diluted with Dulbecco’s phosphate-buffered saline (DPBS; BioWhittaker, Lonza, Belgium). Right after blood collection every citrate blood sample was mixed with 100 μL of their corresponding stock solution to reach the different concentrations of tigecycline. Final concentrations were 0.16 μg/mL for spiking step I, 1.06 μg/mL for step II, 10.6 μg/mL for step III and 106 μg/mL for step IV, respectively. One citrated and one EDTA blood sample served as baseline (i.e., step 0). To exclude diluting effects in control samples 100 μL DPBS was added. After adding the tigecycline solutions, all samples were put aside and slightly swirled for 10 min before ROTEM^®^ measurements were started.

### 4.3. Coagulation Tests

Citrated whole blood samples were centrifuged at 12,500 rpm for 12 min at room temperature. Thereafter further coagulation tests and blood count were performed at the Central Institute of Medical and Chemical Laboratory Diagnostics, University Hospitals of Innsbruck, Austria. For fibrinogen levels (Clauss method) the Multifibren^®^ U-assay (Siemens Healthcare Diagnostics, Marburg, Germany) was used. Fibrinogen antigen was measured using a turbidimetric latex immunoassay (LIAPHEN™ fibrinogen; Hyphen Biomed, Neuville sur Oise, France).

### 4.4. Rotational Thromboelastometry (ROTEM^®^)

All samples were analyzed within 4 h after blood draw. Two ROTEM^®^ gamma analyzers (TEM Innovations, Munich, Germany) with reagents provided by the same manufacturer for EXTEM (extrinsically activated assay with tissue factor), INTEM (intrinsically activated test using ellagic acid), and FIBTEM (extrinsically activated assay with tissue factor and the platelet inhibitor cytochalasin D) measurements were used.

### 4.5. Cell Culture

The human hepatocyte cell line HepG2 (American Type Culture Collection (ATCC^®^) HB-8065TM) was chosen to investigate if tigecycline treatment of cells interferes with fibrinogen production in quantitative and/or qualitative aspects. HepG2 has been tested for an inducible decrease of fibrinogen synthesis in vitro in several studies [24,25,26], mostly reporting fibrinogen concentrations as relative to control experiments or fold changes of mRNA detection. However, Binsack et al. reported fibrinogen concentrations in the supernatant after culturing 6-well plates with 2 mL serum-free medium for 24 h of 0.2 µg/mL = 0.02 mg/dL, which is about 10.000-fold less than normal fibrinogen levels in humans [21].

### 4.6. Stimulation of Hepatocytes

Cells were cultured in Roswell Park Memorial Institute (RPMI 1640, Biochrome GmbH, Berlin, Germany) supplemented with 10% fetal calf serum (FCS) and 1% Penicillin/Streptomycin (Sigma Aldrich, Germany) in collagen I (PureCol, Advanced Biomatrix, San Diego, USA) coated cell culture flasks (25 cm^2^) or 6-well-plates at standard cell culture conditions (5% CO_2_ at 37 °C, saturated humidity). Cells were stimulated with two different concentrations of tigecycline or untreated control cells for 10, 7, 5 days and for 72 and 24 h. Stock preparations of tigecycline were diluted to a physiological concentration (1 µg/mL) and a supraphysiological concentration (100 µg/mL) in PBS and added to the cell culture medium. Medium including tigecycline was exchanged every other day. Cells were checked daily for their confluency status and passaged at a density of 90%. After the respective incubation period, cells were harvested, cell count and viability was assessed and cells were seeded in a standardized density, which equaled 1 × 10^6^ cells per 6-well for fibrinogen measurements and 1 × 10^5^ cells per 96-well to assess the mitochondrial activity. 24 h after seeding, the medium was exchanged to serum-free RPMI. After an additional incubation for 24 h supernatants were harvested, cells and debris were removed by centrifugation.

### 4.7. Real-Time Live Confocal Microscopy of Fibrin Formation

Real-time live confocal microscopy of fibrin formation patterns was performed with a spinning disk confocal system (UltraVIEW^®^ VoX; Perkin Elmer, Waltham, MA, USA) that was connected to a Zeiss AxioObserver Z1 microscope (Zeiss, Oberkochen, Germany). Images and Z-stacks were acquired using Volocity software (Perkin Elmer, Waltham, MA, USA) and a 63x oil immersion objective with a numerical aperture of 1.42 [27,28,29]. Briefly, citrated plasma of healthy volunteers or cell culture supernatants mixed in a 1:1 dilution with plasma from a healthy donor, were pipetted onto a glass slide. Formation of fibrin was initiated by the addition of the ROTEM reagent EXTEM and a protein nicked with fluorescein which specifically labels activated FXIII was added to visualize the fibrin network. To increase the validity of the data acquisition, the experimentator performing the microscopic analysis was blinded to experimental set ups. Background correction was carried out with ImageJ 1.52k (National Institute of Health, USA, http://imageJ.nih.gov/ij), no further adaptions were applied. To validate the observations of the visual examinations, fibrin polymerization of both experimental approaches was quantified by Clauss Method and compared to levels determined by immunologic assays as described above.

### 4.8. Cell Viability and Mitochondrial Activity (MTT)

Cell viability was assessed by acridine orange and propidium iodide staining (Biozym Scientific, Hessisch Oldendorf, Germany) at D0 after trypsinization from 6-well plates and before seeding HepG2 for harvesting. Frequencies of stained cells were determined with an automated cell counter (LUNA, Logos Biosystems, Gyeonggi-do, South Korea).

A nicotinamide adenine dinucleotide phosphate (NAD(P)H)-dependent cellular oxidoreductase enzyme assay, commonly known as MTT assay, was used to assess mitochondrial activity of the cells as proxy for cell viability. Cells were cultured in standardized densities in 96-well plates after priming for the indicated time periods with tigecycline. After 24 h, growth medium was replaced with 200 µL serum-free RPMI per well. After addition of 20 µL MTT reagent (3-[4,5-dimethylthiazol-2-yl]-2,5 diphenyl tetrazolium bromide) cells were incubated for 2 h at standard cell culture conditions. Supernatant was discarded, 200 µL dimethyl sulphoxide (DMSO) was added to each well. After complete disruption of cells the absorbance of formed formazan was read at 570 nm against a reference wavelength at 690 nm with a microplate reader (Infinite M200, Tecan, Salzburg, Austria).

### 4.9. Statistics

The primary endpoint of the study was the absolute change of fibrinogen concentration as dermined by Clauss, from baseline to the measurement after the addition of tigecycline. IBM^®^ SPSS Statistics 25.0 (IBM^®^, Armonk, NY, United States) and GraphPad Prism (Graphpad Software, Version 8.3.0 for macOS, La Jolla, California, USA, www.graphpad.com) were used for statistical analyses and graphical representations. If not else indicated, all parameters are presented as mean and standard deviations.

## Figures and Tables

**Figure 1 antibiotics-09-00084-f001:**
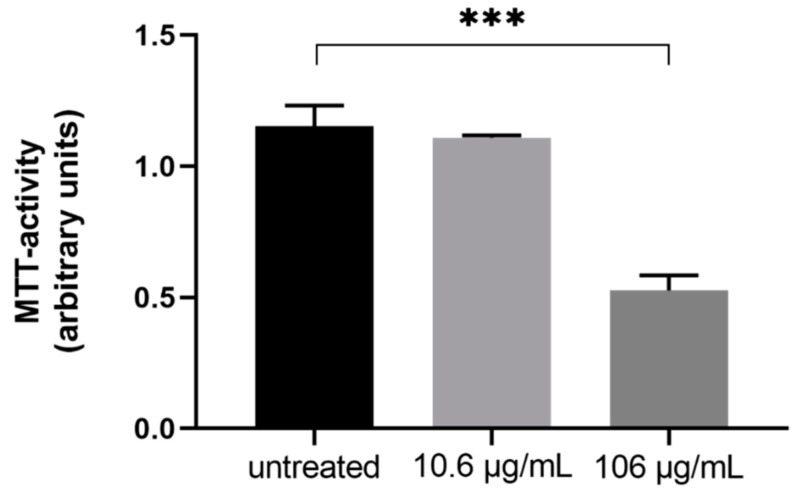
The differences in cell viability as assessed by dimethyl thiazolyl diphenyl tetrazolium bromide (MTT)-assay of immortalized human liver cells (HepG2) after three days treatment with different concentrations of tigecycline (*n* = 3).

**Figure 2 antibiotics-09-00084-f002:**
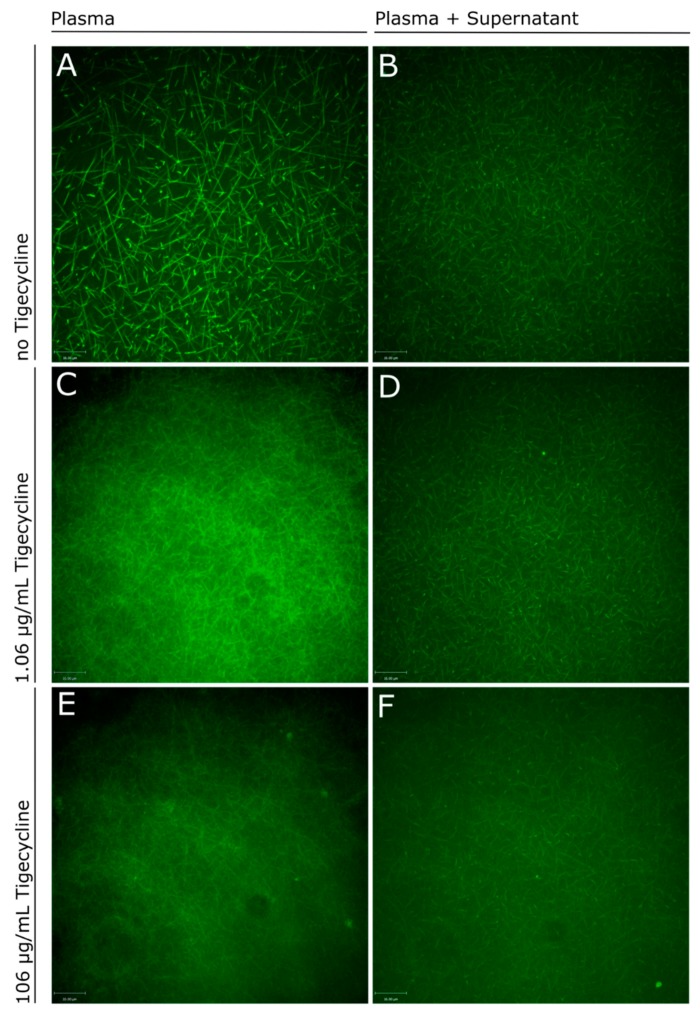
Confocal imaging of fibrin networks in plasma (left panel side) and plasma exchange experiments (right panel side). (**A**) shows the fibrin network in platelet-free plasma (PFP) of a healthy donor and (**B**) mixed with the supernatant of HepG2 cell culture experiments without the addition of tigecycline. Fibrin polymerization was compared to experiments applying increasing doses of tigecycline: PFP mixed with 1.06 µg/mL (**C**) and 106 µg/mL tigecycline (**E**) versus PFP mixed with conditioned supernatant of HepG2 after 24 h incubation in a 1:1 ratio with 1.06 µg/mL (**D**) and 106 µg/mL tigecycline (**F**). green = Fluorescein isothiocyanate (FITC)-labeled FXIIIa; PFP = platelet free plasma.

**Figure 3 antibiotics-09-00084-f003:**
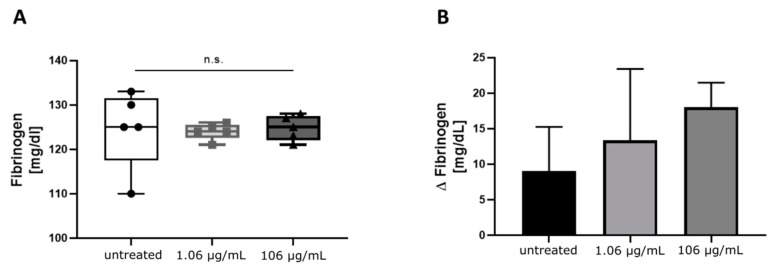
Functional levels of fibrinogen in samples (*n* = 5) of the plasma exchange experiment with cell culture supernatants as determined by Clauss compared to immunologically measured fibrinogen concentrations. (**A**) Fibrinogen levels were detected with Clauss method in supernatants of HepG2 treated with the indicated concentrations of tigecycline and mixed with normal platelet-free plasma (PFP). (**B**) Additional to the functional measurement, antigen levels of fibrinogen were assessed. Plotted are the differences between immunologic minus functional levels of fibrinogen (Δ fibrinogen) as mean and corresponding standard deviation. Both figures display the cumulative data of supernatants after five to ten days incubation of cells with tigecycline plotted by concentration. Differences between groups were tested by One-Way ANOVA, the alpha error levels were higher than 0.05 in both comparisons.

**Table 1 antibiotics-09-00084-t001:** Baseline parameters of all subjects (*n* = 14).

Demographics	Mean ± SD
Age (years)	27.0 ± 6.1
Height (cm)	170.0 ± 9.7
Weight (kg)	64.3 ± 9.6
**Blood count**	
Leukocytes (G/L)	5.6 ± 1.0
Erythrocytes (T/L)	4.6 ± 0.4
Haemoglobin (g/L)	136.2 ± 10.7
Haematocrit (L/L)	0.4 ± 0.0
MCH (pg)	29.9 ± 1.1
MCHC (g/L)	343.5 ± 8.2
MCV (fL)	87.0 ± 3.30
Red blood cell distribution (%)	13.2 ± 0.8
Thrombocytes (G/L)	254.2 ± 51.1
Mean platelet volume (f/L)	10.8 ± 0.81

SD reflects standard deviation, MCH reflects mean corpuscular haemoglobin, MCHC reflects mean corpuscular haemoglobin concentration, MCV reflects mean corpuscular volume.

**Table 2 antibiotics-09-00084-t002:** Standard coagulation parameters after addition of tigecycline in increasing concentrations (*n* = 14).

Tigecycline Concentration (µg/mL)	Baseline	0.106	1.06	10.6	106
aPTT (s)	29.38 ± 2.26	29.38 ± 2.10	29.62 ± 2.33	29.69 ± 2.56	38.92 ± 5.22 ^1^
Quick (%)	91.46 ± 7.32	91.00 ± 6.40	89.46 ± 7.43	90.85 ± 8.29	85.00 ± 5.92
International Normalized Ratio	1.05 ± 0.07	1.05 ± 0.05	1.07 ± 0.06	1.05 ± 0.07	1.09 ± 0.06
Fibrinogen by Clauss (mg/dL)	266.69 ± 60.79	260.10 ± 61.23	258.20 ± 55.29	258.20 ± 49.57	258.10 ± 59.75
Fibrinogen antigen (mg/dL)	258.23 ± 34.47	245.20 ± 33.19	244.30 ± 33.73	241.50 ± 39.00	247.80 ± 34.76
Thrombin time (s)	20.38 ± 1.76	19.92 ± 1.80	20.15 ± 1.73	19.77 ± 1.88	20.38 ± 1.98
Batroxobin time (s)	16.38 ± 0.51	16.31 ± 0.75	16.23 ± 0.60	16.15 ± 0.55	16.23 ± 0.44
F. XIII (%)	108.44 ± 13.93	114.30 ± 16.06	113.70 ± 17.85	114.20 ± 15.85	112.20 ± 16.15
F. XIII, immunologic (%)	103.18 ± 9.23	106.10 ± 10.97	105.50 ± 10.42	105.90 ± 8.93	105.60 ± 9.12

^1^*p* < 0.0001. aPTT reflects activated partial thromboplastin time, F. XIII reflects coagulation factor XIII, F. XIII, immunologic reflects coagulation factor XIII as assessed immunologically. Data are presented as means ± standard deviations; Statistically significant differences between groups were tested by One-Way ANOVA or Kruskal–Wallis-Test. Post-hoc multiple comparison of significant results were tested by Dunnet’s multiple comparison test.

**Table 3 antibiotics-09-00084-t003:** Rotational thromboelastometry after addition of tigecycline in increasing concentrations in blood samples of healthy volunteers (*n* = 14).

Tigecycline Concentration (µg/mL)	Baseline	0.106	1.06	10.6	106
**INTEM**					
Clotting Time (s)	189.3 ± 19.4	194 ± 39	194 ± 27	182 ± 54	192 ± 22
Maximum Clot Firmness (mm)	56.5 ± 4.3	58 ± 5	57 ± 5	57 ± 5	58 ± 5
Maximum Lysis (%)	11.0 ± 2.2	10 ± 2	9 ± 2	10 ± 2	11 ± 2
**EXTEM**					
Clotting Time (s)	75.0 ± 9.2	76 ± 10	72 ± 6	72 ± 11	79 ± 12
Maximum Clot Firmness (mm)	61.2 ± 5.6	61 ± 6	59 ± 6	60 ± 6	61 ± 7
Maximum Lysis (%)	9.2 ± 1.9	9 ± 2	9 ± 2	9 ± 2	9 ± 3
**FIBTEM**					
Clotting Time (s)	63.8 ± 16.3	71 ± 7	68 ± 17	67 ± 10	72 ± 9
Maximum Clot Firmness (mm)	13.9 ± 5.3	11 ± 4	13 ± 2	15 ± 6	12 ± 2
Maximum Lysis (%)	3.6 ± 5.4	2 ± 4	4 ± 3	9 ± 11	2 ± 3

INTEM reflects intrinsically activated test using ellagic acid, EXTEM reflects extrinsically activated assay with tissue factor, FIBTEM reflects extrinsically activated assay with tissue factor and the platelet inhibitor cytochalasin D. Data are presented as means ± standard deviations.

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
