# Peer review of "Tigecycline Interferes with Fibrinogen Polymerization Independent of Peripheral Interactions with the Coagulation System"

_antibiotics, 2020, doi:10.3390/antibiotics9020084_

Round 1
Reviewer 1 Report
1. Brandtner et al., described presented a well-written manuscript on the Tigecycline effect on Fibrinogen polymerization. It is an interesting research objective to pursue, however, there are significant downsides with reported work.
In my opinion, authors should try with patient samples rather than healthy samples to identify any crucial observations. I strongly believe that in addition to tigecycline dose, infection conditions are very key for this observed side effects. At least the authors should compare the current reported data with patient samples' data in key experiments. Rewriting of figure -2 caption would be needed. It is misleading and hard to follow. If I understood it correctly, I do see only five data points (N = 5) in figure 3. Based on the earlier description, it is assumed N = 14 was used in all the assays. In all figure and table captions, the sample size should be mentioned. More assays might be needed to assess the impaired hepatic synthesis of coagulation factors. MTT assay may not be the ideal one.
Author Response
Point 1: In my opinion, authors should try with patient samples rather than healthy samples to identify any crucial observations. I strongly believe that in addition to tigecycline dose, infection conditions are very key for this observed side effects. At least the authors should compare the current reported data with patient samples' data in key experiments.
Response 1. The reviewer makes an important point. At this (first) stage we sought to exclude a false low measurement of fibrinogen during tigecycline. From our clinical observations only one third of severely ill patients experience a decrease of fibrinogen levels during tigecycline treatment. Moreover, clinical relevant bleeding seems to be extremely rare in these patients. To farther clarify the underlying reason, we currently conduct a retrospective examination of our critical ill patients receiving tigecycline in the last few years (ongoing data collection).
Point 2. Rewriting of figure -2 caption would be needed. It is misleading and hard to follow.
Response 2. The reviewer correctly states this point, the caption has been changed correspondingly (see page 6, line 137ff).
Point 3. If I understood it correctly, I do see only five data points (N = 5) in figure 3. Based on the earlier description, it is assumed N = 14 was used in all the assays. In all figure and table captions, the sample size should be mentioned.
Response 3. The reviewer rightly criticises the figure and table descriptions. The number of subjects was been included in captions of figures and table (see page 3, line 93 and line 103; page 4, line 115; page 5, line 126; page 8, line 153).
The number of data points in figure 3 refers to the in-vitro experiments, where HepG2 was incubated with different doses of tigecycline. For this assay the number of experiments per concentration was 5. We rephrased the reporting of the results (page 7, line 145ff) to clarify the issue.
Point 4: More assays might be needed to assess the impaired hepatic synthesis of coagulation factors. MTT assay may not be the ideal one.
Response 4. We agree that further analyses and different methods would be necessary to investigate the hepatic impairment of coagulation factor synthesis in tigecycline treatment. The purpose of showing the MTT-results of the in-vitro experiments is to display, that physiologic doses did not have a toxic effect on hepatocytes, while supraphysiologic doses did. However, this had no impact on functional fibrinogen quantification (figure 3a). To better elaborate the connection between these assays, we added this consideration to the results section describing figure 3a (page 7, line 147ff).
Reviewer 2 Report
Content suggestions:
The authors used a small cohort for the study – this may be the reason for obtaining of false results (bias). The authors tested a group of healthy volunteers using tigecycline instead of the patients. For the comparison, they could test both groups of individuals with the results more useful for the practice. If the authors tested also the group of the patients in which tigecycline is indicated, they could correlate its pharmacokinetics with the markers of the sepsis / inflammation (CRP, IL-6…). Did the authors use the hepatoprotective drugs to protect the liver from the effects of tigecycline when confirming a rapid loss of mitochondrial activity in hepatic cells with supra-therapeutic tigecycline dosages ? Did the authors took into account the effect of other factors on the liver function tests (the analgetic drugs, steatosis – the effect of an unhealthy lifestyle...) ? What are the recommendations of the authors based on the results for the clinical practice ?
The article could be edited after minor revision according to comments to the authors.
Author Response
Point 1: The authors used a small cohort for the study – this may be the reason for obtaining of false results (bias).
Response 1: The reviewer correctly adds a possible limitation, which has been added (see page 8, line 197ff).
Point 2: The authors tested a group of healthy volunteers using tigecycline instead of the patients. For the comparison, they could test both groups of individuals with the results more useful for the practice. If the authors tested also the group of the patients in which tigecycline is indicated, they could correlate its pharmacokinetics with the markers of the sepsis / inflammation (CRP, IL-6…).
Response 2: The reviewer makes an important point. The healthy subjects in our investigation lack the severe infections of patients receiving tigecycline. To further clarify the underlying reason, we currently conduct a retrospective examination of our critical ill patients receiving tigecycline (ongoing data collection).
Point 3: Did the authors use the hepatoprotective drugs to protect the liver from the effects of tigecycline when confirming a rapid loss of mitochondrial activity in hepatic cells with supra-therapeutic tigecycline dosages?
Response 3: The reviewer rightly considers using of hepatoprotective drugs like L-carnitine, Vitamin C or N-acetylcysteine to confirm a rapid decrease of mitochondrial activity in hepatic cells with supra-therapeutic tigecycline dosages. However, the hepatotoxic effect in-vitro did not influence functional fibrinogen measurements in plasma exchange experiments of supernatants (figure 3A). Thus, we did not attribute the effect of tigecycline on hepatic cells to the observed qualitative changes in fibrin polymerizations (figure 2).
Clearly, addition of hepatoprotective drugs should be part of further investigating a possible hepatotoxic effect of tigecycline. We have added this point to the discussion section (see page 7, line 179f).
Point 4: Did the authors took into account the effect of other factors on the liver function tests (the analgetic drugs, steatosis – the effect of an unhealthy lifestyle...)?
Response 4: The reviewer rightly questions possible effects on liver function tests like several over-the-counter drugs or an unhealthy lifestyle. As part of the medical check-up before blood sampling all subjects negated “liver problems” and “ample intake of alcohol (> 20 g alcohol in male or > 15 g alcohol in female subjects at the day before sampling). An unhealthy lifestyle seemed to be unlikely in our healthy subjects, but cannot be ruled out. Moreover, taken medications within one month before blood sampling were recorded. The materials section has been changed accordingly (see page 8, line 216ff).
Point 5: What are the recommendations of the authors based on the results for the clinical practice?
Response 5: The reviewer correctly states missing recommendations for the clinical practice. We have changed the discussion section accordingly (page 8, line 202ff).
Point 6: The article could be edited after minor revision according to comments to the authors.
Response 6: We thank the reviewer for the excellent comments and hope to have improved the manuscript accordingly.
Round 2
Reviewer 1 Report
As the authors addressed the majority of the concerns and comments, the manuscript can be published in the present format.